# Diminished Immune Response and Elevated Abundance in Gut Microbe Dubosiella in Mouse Models of Chronic Colitis with GBP5 Deficiency

**DOI:** 10.3390/biom14070873

**Published:** 2024-07-20

**Authors:** Yichen Li, Wenxia Wang, Yuxuan Liu, Senru Li, Jingyu Wang, Linlin Hou

**Affiliations:** 1Medical College, Jiaying University, Meizhou 514031, China; 2Department of Immunology and Microbiology, Zhongshan School of Medicine, Sun Yat-sen University, Guangzhou 510080, China; wangwx63@mail2.sysu.edu.cn; 3Guangdong Institute of Gastroenterology, Guangdong Provincial Key Laboratory of Colorectal and Pelvic Floor Diseases, Biomedical Innovation Center, Department of General Surgery, The Six Affiliated Hospital, Sun Yat-sen University, Guangzhou 510655, China; 4School of Medicine, Shenzhen Campus of Sun Yat-sen University, Shenzhen 518107, China; liuyx355@mail2.sysu.edu.cn (Y.L.); lisr23@mail2.sysu.edu.cn (S.L.); wangjy363@mail2.sysu.edu.cn (J.W.)

**Keywords:** inflammatory bowel disease, chronic colitis, guanylate binding protein, gut microbiota

## Abstract

Guanylate binding protein 5 (GBP5) is an emerging immune component that has been increasingly recognized for its involvement in autoimmune diseases, particularly inflammatory bowel disease (IBD). IBD is a complex disease involving inflammation of the gastrointestinal tract. Here, we explored the functional significance of GBP5 using *Gbp5* knockout mice and wildtype mice exposed to dextran sulfate sodium (DSS) to generate chronic colitis model. We found that *Gbp5* deficiency protected mice from DSS-induced chronic colitis. Transcriptome analysis of colon tissues showed reduced immune responses in *Gbp5* knockout mice compared to those in corresponding wildtype mice. We further observed that after repeated DSS exposure, the gut microbiota was altered, both in wildtype mice and *Gbp5* knockout mice; however, the gut microbiome health index was higher in the *Gbp5* knockout mice. Notably, a probiotic murine commensal bacterium, *Dubosiella*, was predominantly enriched in these knockout mice. Our findings suggest that GBP5 plays an important role in promoting inflammation and dysbiosis in the intestine, the prevention of which might therefore be worth exploring in regards to IBD treatment.

## 1. Introduction

Inflammatory bowel disease (IBD) is defined as an autoimmune disease which affects the intestine and multiple systems. The pathogenesis of IBD is multifactorial, and factors such as genetics, diet, gut microbiota dysbiosis, and environmental influences all contributing to its development [1]. The etiology of IBD is still unclear, but the disease appears to occur in genetically susceptible individuals by an dysregulation of both innate and adaptive immune responses against intestinal microorganisms [2]. The composition of the gut microbiota differs between healthy participants and IBD patients. Patients with IBD exhibit a decrease in α diversity, *Bacteroides*, *Firmicutes*, *Clostridia*, *Ruminococcaceae*, *Bifidobacterium*, *Lactobacillus*, and *Faecalibacterium prausnitzii*, as well as an increase in *Gammaproteobacteria*, *Escherichia coli*, and *Fusobacterium* species [3,4]. Several reports have indicated that certain probiotic bacterial strains, such as *Escherichia coli Nissle* 1917, *Lactobacillus rhamnosus* GG, and *Enterobacter ludwigii*, improved IBD or mouse models of colitis [5,6,7].

Guanylate binding proteins (GBPs) are a family of GTPases highly expressed in human and murine cells exposed to interferon γ (IFNγ) [8]. Seven human GBPs are encoded on chromosome 1, while 11 mouse GBPs are encoded on chromosome 3 and chromosome 5 [9,10]. GBPs are central players in host defense against bacterial, viral, and protozoan pathogens [11,12,13]. GBP5, a founding member of the GBP family, plays an emerging role in mediating defenses against intracellular pathogens, such as *Francisella novicida*, HIV-1 virus, influenza A virus, and *Toxoplasma gondii* [14,15,16,17]. Published studies suggest that GBP5 plays a role in innate immunity and inflammation by activating NLRP3 and AIM2 inflammasomes [18,19].

Our previous study revealed that GBP5 is highly expressed in the inflamed colons of IBD patients, and that GBP5 is required for the stimulated production of a large number of cytokines and chemokines in vitro, suggesting a pro-inflammatory role of GBP5 in IBD [20,21]. Recently, Luu et.al found that the downregulation of GBP5 is associated with disease remission following fecal microbiota transplantation in IBD patients [22]. In this work, we show that *Gbp5* deficiency protects mice from dextran sulfate sodium (DSS)-induced chronic colitis. The absence of *Gbp5* downregulated the immune response associated signaling pathways and enriched the probiotic bacterial statin *Dubosiella*. These results provide new insight into the role of GBP5 in immune regulation and changing the composition of gut microbiota in vivo.

## 2. Materials and Methods

### 2.1. Mice

*Gbp5*^+/−^ mice in C57BL/6J were purchased from the GemPharmatech Co., Ltd. (Jiangsu, China). Heterozygous mice were mated to generate homozygous offsprings. Genome types of +/+, +/−, and −/− mice were determined by PCR, according to the instructions of the supplier. The mice were maintained under specific pathogen free (SPF) conditions at the Sixth Affiliated Hospital of Sun Yat-sen University. The mice studies were carried out following the guidelines of the Institutional Animal Care and Use Committee of the Sixth Affiliated Hospital of Sun Yat-sen University (approval number: IACUC-2023022201, approved on 22 February 2023). 

### 2.2. Dextran Sodium Sulfate (DSS)-Induced Chronic Colitis

For DSS-induced chronic colitis, 6–8-week-old male mice were used. To select an appropriate DSS concentration, pre-experiments were performed. It was found that 2.5% (*w*/*v*) DSS allowed for a successful model of chronic colitis. In contrast, a lower concentration of DSS was insufficient to induce inflammation, while a higher concentration led to excessive death of the mice. The *Gbp5*^−/−^ mice or wildtype (WT) littermates were randomly divided into two groups. Each group consisted of five age- and sex-matched mice. One group of mice received oral administration of 2.5% DSS (Meilunbio, Dalian, China), dissolved in drinking water, for one week, then sterile water for 2 weeks, for a total of three cycles in sequence, while the other group consumed sterile water only. The mice were sacrificed on day 63. The colon tissues were fixed in 4% paraformaldehyde for H&E staining. The disease activity index (DAI) [23], histological score [24,25], macroscopic score [26], and microscopic score [27] were calculated as previous reported, as briefly explained in Appendix A.

### 2.3. Quantitative Real-Time PCR (RT-qPCR) Analysis

RNA from the colon tissues of WT mice, untreated or treated with DSS, were isolated using TRIzol (Invitrogen, Carlsbad, CA, USA) and converted into cDNA using Fast Reverse Transcription kits (ES Science, Shanghai, China), according to the manufacturer’s instructions. The RT-qPCR method was employed as follows: 95 °C for 30 s, followed by 40 cycles of 95 °C for 10 s, and 60 °C for 30 s. The RT-qPCR results were analyzed using the relative quantitative 2^−ΔΔCT^ method. Gene expression was assessed using the 2× SYBR mix, with the primers as follows: 

mouse *Gbp5*-forward: 5′-TTCACCCAATCTAAGACCAAGAC-3′;

mouse *Gbp5*-reverse: 5′-AGCACCAGGCTTTCTAGACG-3′; 

mouse *Ifng*-forward: 5′-GCCACGGCACAGTCATTGA-3′

mouse *Ifng*-reverse: 5′-TGCTGATGGCCTGATTGTCTT-3′

mouse *Actb*-forward: 5′-ACCTGACAGACTACCTCATGAAGA-3′

mouse *Actb*-reverse: 5′-TCATGGATGCCACAGGATTCCATA-3′

### 2.4. Luminex Bead-Based Multiplex Assay

The colon tissues and serums of the mice were collected and subjected to Luminex bead-based multiplex assay. The tissue samples were lysed, and the proteins were quantitated using a standard BCA protocol. Finally, the equal mass of the tissues (45 μg) or the equal volume of the serums (50 μL) were loaded. The assay was conducted according, to the manufacturer’s protocol, using Bio-Plex Pro Mouse Cytokine Grp I Panel 23-plex (#M60009RDPD) with the Luminex 200 system from Wayen Biotechnologies (Shanghai, China).

### 2.5. Transcriptome Analysis

Total RNA was extracted from the tissue using TRIzol, according the manufacturer’s instructions. Only high-quality RNA samples were used to construct the sequencing library. RNA purification, reverse transcription, library construction, and sequencing were performed at Shanghai Majorbio Bio-pharm Biotechnology Co., Ltd. (Shanghai, China), according to the manufacturer’s instructions (Illumina, San Diego, CA, USA). The RNA-seq transcriptome library was prepared using Illumina^®^ Stranded mRNA Prep, Ligation from Illumina (San Diego, CA, USA), employing 1 μg of total RNA. 

The raw paired end reads were trimmed and quality controlled using fastp v0.23.4 [28], with default parameters. Then, clean reads were separately aligned to the reference genome in the orientation mode using HISAT2 v2.1 software [29]. The mapped reads of each sample were assembled by StringTie [30] using a reference-based approach. To identify DEGs (differential expression genes) between two different samples, the expression level of each transcript was calculated according to the transcripts per million reads (TPM) method. RSEM v1.3.3 [31] was used to quantify gene abundances. Essentially, differential expression analysis was performed using DESeq2 v1.44.0 [32]. DEGs with |log_2_FC| >= 1 and FDR < 0.05 were considered to be significantly different expressed genes. All data were analyzed using the Majorbio Cloud Platform (https://cloud.majorbio.com/, accessed on 7 March 2024).

### 2.6. Microbiome Analysis

Fecal samples were collected and stored immediately at −80 °C. Bacteria DNA was isolated using the CTAB/SDS method at Majorbio Bio-pharm Biotechnology Co., Ltd. (Shanghai, China). DNA concentration and purity were monitored on 1% agarose gels. Depending to the concentration, DNA was diluted to 1 ng/μL using sterile water. 

The 16S rRNA genes of distinct regions were then amplified with the barcode, using specific primers. All PCR reactions were carried out using TransStart^®^ FastPfu DNA Polymerase (TransGen Biotech, Beijing, China). Sequencing libraries were generated using the SMRTbellTM Template Prep Kit (PacBio, Menlo Park, CA, USA). Library quality was assessed on the Qubit@ 2.0 Fluorometer (ThermoFisher Scientific, Waltham, MA, USA) and the FEMTO Pulse system. The library was then sequenced on the PacBio Sequel platform. 

Raw sequences were initially processed through the PacBio SMRT portal. Sequences were filtered for a minimum of three passes, and a predicted accuracy of at least 90% (minfullpass = 3, min predicted accuracy = 0.9). The files generated by the PacBio platform were then used for amplicon size trimming to remove sequences outside the expected amplicon size. The reads were compared with the reference database using the UCHIME algorithm (http://www.drive5.com/usearch/manual/uchime_algo.html, accessed on 14 December 2023) to detect chimera sequences, and the chimera sequences were then removed [33,34], after which the clean reads were finally obtained. Sequence analyses were performed using Uparse software (Uparse v7.0.1001, http://drive5.com/uparse/, accessed on 14 December 2023) [35]. 

For each representative sequence, taxonomic information was annotated using the SSUrRNA data from the Silva database (https://www.arb-silva.de/, accessed on 14 December 2023) based on the Mothur algorithm [36,37]. To investigate the phylogenetic relationships of the different OTUs and the differences in the dominant species in different samples, multiple sequence alignments were conducted using MUSCLE software (Version 3.8.31, http://www.drive5.com/muscle/, accessed on 14 December 2023) [38]. The linear discriminant analysis effect size (LEfSe) method [39] was used to determine discriminatory taxa for each group. The microbial dysbiosis index (MDI) is defined as the log_10_ of the [total abundance in organisms increased in CD] over the [total abundance of the organisms decreased in CD], as previously reported [40]. The gut microbiome health index (GMHI), formulated based on 50 microbial species associated with healthy gut ecosystems, was used for evaluating the health status of the microbiota [41].

### 2.7. Statistical Analysis

Data were analyzed using either paired or unpaired two-tailed Student’s *t*-test with GraphPad Prism (v8.0). The statistical methods used in each experiment were reported in the figure legends. Data were represented as the mean ± SEM (standard error of the mean). The correlation was evaluated using Spearman’s correlation. *p* < 0.05 was considered to be statistically significant. 

## 3. Results

### 3.1. Gbp5 Deficiency Alleviated the Symptoms of DSS-Induced Chronic Colitis in Mice

Wildtype (WT) and *Gbp5* knockout mice (*Gbp5*^−/−^) were repetitively treated with DSS (control mice drinking clean water) for 9 weeks to generate experimental chronic colitis (Figure 1A). In the control groups, the accelerated body weight gain rates of *Gbp5*^−/−^ mice were observed during this period; moreover, DSS-treated *Gbp5*^−/−^ mice exhibited a significant reduction in loss of body weight compared to the corresponding WT mice (Figure 1B). By normalizing the weight of each mouse to the initial weight in every DSS cycle, the weight loss of DSS treated mice was affected to a lesser extent in *Gbp5*^−/−^ compared to the WT mice (Figure 1C). After DSS treatment, the expression of *Gbp5* in the colons was elevated, and there was strong positive correlation between *Gbp5* and *Ifng* expression (Figure 1D). The mice were euthanized at day 63 to evaluate the severity of the disease, and the disease activity index (DAI) was reduced after chronic DSS administration in *Gbp5*^−/−^ mice (Figure 1E). There was no difference regarding colon length between WT and *Gbp5*^−/−^ mice in the control groups, but a longer colon length was observed in the DSS-treated *Gbp5*^−/−^ mice compared to that in the corresponding WT mice (Figure 1F). These results showed that the DSS-induced *Gbp5* in the inflamed colon played a crucial role in chronic colitis.

### 3.2. Loss of Gbp5 Decreased Inflammation in Chronic Colitis

The colon tissue of WT and *Gbp5*^−/−^ mice, untreated or treated with three cycles DSS, were subjected to hematoxylin and eosin (H&E) staining; there was no difference in histology between WT and *Gbp5*^−/−^ mice in the control groups. DSS administration induced inflammation both in WT and *Gbp5*^−/−^ mice, but *Gbp5*^−/−^ mice presented a significantly lower inflammatory cell infiltration (Figure 2A). In addition, the loss of *Gbp5* mitigated colitis compared with its severity in the WT littermate controls, with a significantly decreased histological score, macroscopic score, and microscopic score (Figure 2B–D). To further explore the effect of *Gbp5* deficiency on cytokines expression, the colon tissues and serums of the mice were subjected to Luminex assay for quantification of 23 cytokines. The results showed that, the levels of 10 cytokines, including Eotaxin (CCL11), G-CSF, IFN-γ, IL-6, TNFα, IL-17A, IL-1β, KC (CXCL1), MCP-1 (CCL2), and MIP-1α in the colon were lower in *Gbp5*^−/−^ mice treated with DSS compared to the levels in the corresponding WT mice (Figure 2E, Appendix A). There were no differences in cytokine expression in the serum between these two groups (Figure 2E, Appendix A). These findings suggest that *Gbp5* deficiency inhibited the stimulation of pro-inflammatory cytokines in the gut and ameliorated DSS-induced inflammation in chronic colitis.

### 3.3. Immune Response-Associated Pathways Were Downregulated by DSS Exposure in the Gbp5^−/−^ Mice

To reveal the underlying mechanisms of GBP5 in regards to inflammation, RNA sequencing (RNA-seq) analysis was performed on colon tissue of WT and *Gbp5*^−/−^ mice. The principal component analysis (PCA) revealed the distinct clustering of transcriptomic profiles between groups (WT, *Gbp5*^−/−^, WT_DSS, and *Gbp5*^−/−^_DSS) (Figure 3A). A total of 219 DEGs (154 down- and 65 upregulated genes) were screened out of the *Gbp5*^−/−^_DSS group compared to the WT_DSS group (Figure 3B). DEGs were then annotated into biological process (BP), cellular component (CC), and molecular function (MF) using GO annotation analysis and KEGG enrichment analysis. The top five enriched pathways are presented in every item; notably, the degree of enrichment (Z score) for these pathways was lower in *Gbp5*^−/−^_DSS than in WT_DSS (Figure 3C). Most of these pathways were immune response-associated, such as lymphocyte mediated immunity; adaptive immune response based on somatic recombination of immune receptors built from immunoglobulin superfamily domains; and the defense response to bacterium and the immunoglobulin complex, antigen binding, cytokine–cytokine receptor interaction, and the IL-17 signaling pathway. These data indicating that the DSS-induced colonic immune response, particularly the innate antimicrobial immune response, was reduced with *Gbp5* knockout. 

### 3.4. Gbp5 Deficiency Alteration of the Gut Microbiota in Mice

From the gene-pathway analysis of our RNA-seq data between the WT_DSS and *Gbp5*^−/−^_DSS mice, we noticed a significant enrichment of genes involved in bacterial infection. To further characterize *Gbp5* deficiency-induced alterations in microbiota, the gut microbiota compositions of WT and *Gbp5*^−/−^ mice were determined before and after three cycles of DSS administration using full-length 16S ribosomal DNA sequencing analysis (Figure 4A). There was no difference in microbial alpha diversity between the pretreated WT and *Gbp5*^−/−^ mice, but it increased in *Gbp5*^−/−^ mice after DSS treatment compared to that noted in the corresponding WT mice (a significant difference was only observed in the “ace” index) (Figure 4B,C). Taxonomic analysis of the microbiome using principal coordinate analysis (PCoA) showed that mice treated with DSS exhibited significant clustering separation of the microbiome compared with that of those in the pretreated groups, but there were still differences in the gut bacterial composition between post-treated WT and *Gbp5*^−/−^ mice (Figure 4D). The Venn diagram also shows that the composition and variety of the gut microbiota changed in WT and *Gbp5*^−/−^ mice, untreated or treated with DSS (Figure 4E).

### 3.5. Gbp5 Deficiency Regulates Colon Microbial Homeostasis and Enriches Dubosiella 

The gut microbial composition at the phylum level in post-treated and pre-treated mice showed significant differences, especially in regards to the increased abundance of Verrucomicrobiota in the DSS treated mice (Figure 5A). By analyzing the gut microbial composition at the genus level between post-treated WT and *Gbp5*^−/−^ mice, we observed increasing amounts of *Akkemansia* and *norank_f_Muribaculaceae*, and decreasing amounts of *Lactobacillus*, *Turicibacter*, and *Faecalibaculum* in *Gbp5*^−/−^ mice (Figure 5B). We then used the microbial dysbiosis index (MDI) to evaluate the severity of gut microbiota dysbiosis and found that the trend of MDI was reduced in post-treated *Gbp5*^−/−^ mice compared to that in the corresponding WT mice (Figure 5C). Taking this observation a step further, we also found that the gut microbiome health index (GMHI) was increased in post-treated *Gbp5*^−/−^ mice compared to that for the corresponding WT mice (Figure 5D). A linear discriminant analysis of effect size (LEfSe) was used to detect marked differences in the predominance of bacterial communities between post-treated WT and *Gbp5*^−/−^ mice. Notably, *Gbp5* deficiency induced the accumulation of *Dubosiella*, *Candidatus_Saccharimonas*, *norank_f_Erysipelotrichaceae* and ASF356 (Figure 5E). However, using the Student’s *t*-test, we found that *Dubosiella* was predominantly enriched in post-treated *Gbp5*^−/−^ mice compared to the corresponding WT mice (Figure 5F). Taken together, upon DSS exposure, loss of *Gbp5* led to an enrichment of *Dubosiella,* a probiotic murine commensal bacterium, suggesting that DSS-induced upregulation of *Gbp5* affected the gut microbial homeostasis.

## 4. Discussion

This study demonstrates that *Gbp5* deficiency reduces susceptibility to DSS-induced chronic colitis. Loss of *Gbp5* reduced intestinal inflammation and immune response. Moreover, *Gbp5* deficiency alters the gut microbiota and enriches *Dubosiella*. This sheds light on the interaction between bacteria and the immune system in the inflamed intestine.

IBD presents two major clinical phenotypes: Crohn’s disease (CD) and ulcerative colitis (UC). CD can involve any part of the gastrointestinal tract, from mouth to anus, whereas UC primarily involves confluent inflammation of the colonic mucosa [42]. The DSS-induced chronic colitis mouse model closely resembles human UC [43]. Our previous study showed that GBP5 drives more intense innate immunity and inflammatory responses in human CD than those in UC [44], indicating that GBP5 plays an important role in the pathogenesis of both CD and UC. Consequently, the outcomes of the UC animal model established in this study may also serve as a valuable reference for CD.

Inflammatory factors such IFN-γ, TNF-α, and IL-1β can induce the expression of GBP5 [45,46]; these cytokines are highly expressed in IBD and in the experimental mouse models of colitis [47]. Our previous study showed that the induction of various cytokines, including IFN-γ, TNF-α, and IL-1β, was GBP5-dependent in the THP-1 cells [21]. In this study, it was revealed that upon DSS exposure, *Gbp5* deficiency decreased eotaxin, G-CSF, IFN-γ, IL-6, TNFα, IL-17A, IL-1β, KC, MCP-1, and MIP-1α induction in the colon tissue of mice, and did not have a significant effect on these cytokines in the blood. Although the regulatory mechanism of GBP5 in regards to cytokines remains unclear, our findings have broadened the understanding of its in vivo functions. 

The abnormal immune response in IBD is associated with dysregulation of both innate and adaptive immunity. An impaired innate immunity results in the loss of regulation over an altered intestinal microbiota and triggers the activation of the adaptive immune system, thereby promoting a secondary inflammatory response that is responsible for tissue damage in IBD [48,49]. Based on the GO analysis, we demonstrated the pathways related to immune response, i.e., lymphocyte mediated immunity, positive regulation of lymphocyte activation, adaptive immune response based on somatic recombination of immune receptors built from immunoglobulin superfamily domains, and humoral immune response, were all downregulated after *Gbp5* knockout. The activation and homing of lymphocytes, including B cells, T cells, regulatory T cells, mucosal T cells, T helper cells, and cytotoxic T cells, in inflamed intestine are key phenomena in IBD [50]. Preventing lymphocytes trafficking from the circulation into the gut tissue is considered an efficacious therapeutic option for IBD patients [51]. Based on KEGG enrichment, we also discovered that the gene sets related to cytokine-cytokine receptor interaction, inflammatory bowel disease, rheumatoid arthritis, and the IL-17 signaling pathway were downregulated in *Gbp5*^−/−^ mice. From the gene-pathway analysis of our transcriptome data comparing *Gbp5*^−/−^ mice exposed to DSS and WT mice exposed to DSS, we noticed a significant enrichment of genes involved in the defense response to bacterium and viral protein interaction with cytokine and cytokine receptor. GBP5 is a major factor in mediating defenses against intracellular pathogens; however, our study was conducted on specific pathogen-free animals which were not infected with exogenous pathogenic microorganisms. These results highlight the importance of GBP5 in regards to adaptive immunity in the DSS model of colitis. 

T helper (Th) 17 cells are a major component of CD4^+^ T cells, specifically producing IL-17 [52]. Th17 cells and IL-17 play crucial roles in the occurrence and development of IBD and serve as a bridge between gut microbes and the gut immune system [53]. A prominent characteristic of IBD is the decrease in beneficial bacteria and the increase in pathogenic bacteria [54]. The commensal bacteria degrade food in the intestine to supply energy to cells, generate short chain fatty acids (SCFAs) to maintain gut homeostasis, and drive host resilience to pathogen invasion [55,56]. A study identified a probiotic murine commensal bacterium, *Dubosiella*, that ameliorates DSS-induced colitis by rebalancing Treg/Th17 responses and improving mucosal barrier integrity [57]. *Dubosiella* was reported to have an anti-aging function and acts as a crucial genus for enhancing exercise effectiveness in the treatment of NAFLD [58,59]. From these previous studies, *Dubosiella* may represent a promising therapeutic target for diseases accompanied by dysbiosis. In line with the results of these studies, our results showed that *Dubosiella* was accumulated in *Gbp5*^−/−^ mice treated with DSS compared to the results for corresponding WT mice, which indicates that GBP5 has a role in controlling intestinal homeostasis.

The major limitation of this study is that although we have found that GBP5 can promote the occurrence and development of colonic inflammation in vivo, the specific mechanism of how GBP5 exacerbates the immune response has not been fully determined. In addition, GBP5 as a molecule that helps cells produce autonomous immunity against pathogenic microorganisms, as well as its effect on the structure of microbiota in the complex and diverse microbial environment of the intestine, requires further study. Moreover, the mice used in this study were bred and hosted under specific pathogen-free conditions which strongly influences gut microbiota; therefore, the results gathered under conventional conditions or in the case of human disease could be different. 

## 5. Conclusions

Currently, there are no therapies focused on GBPs in autoimmune diseases. Thus, this area presents an exciting opportunity, given the putative role of GBP5 as an endogenous regulator of immune response and gut microbiota.

## Figures and Tables

**Figure 1 biomolecules-14-00873-f001:**
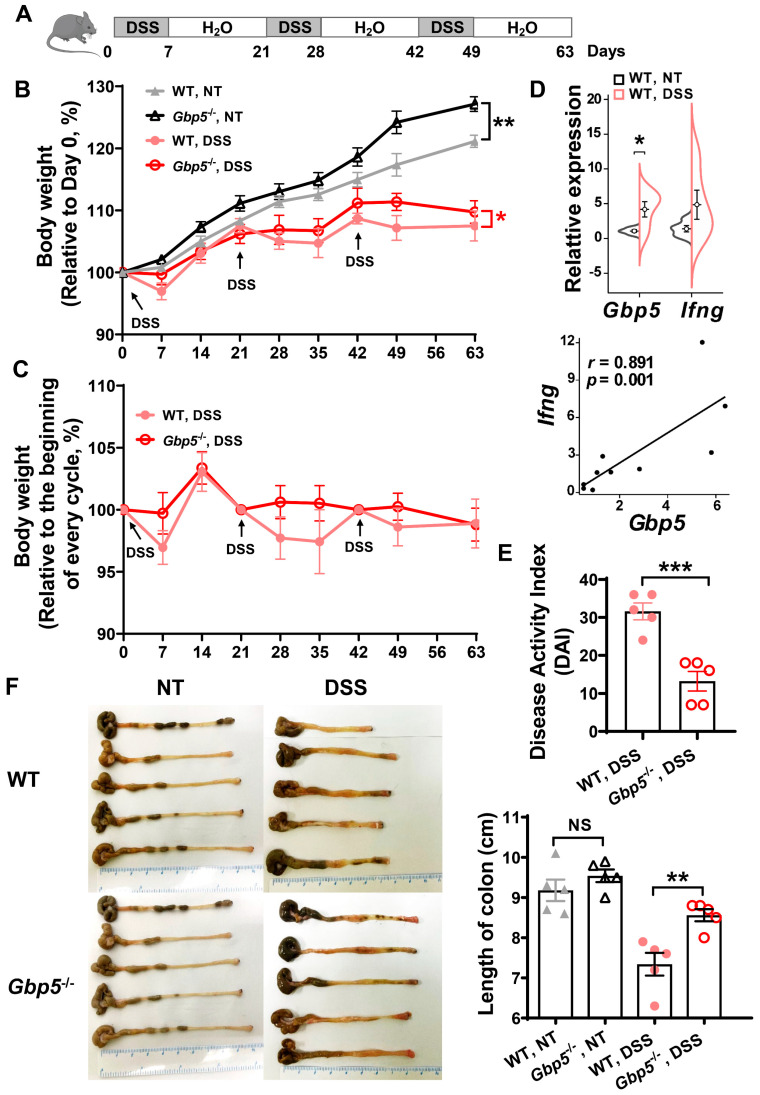
*Gbp5* knockout attenuates DSS-induced chronic colitis. (**A**) Experimental scheme for DSS-induced chronic colitis. DSS water (2.5%) was repetitively administered to mice to induce colitis. (**B**) The percentage of body weight change normalized by the initial weight of each mouse. Paired Student’s *t*-test, *, *p* < 0.05. (**C**) The percentage of body weight change normalized by the initial weight of each mouse in every DSS cycle. (**D**) The mRNA level and Spearman’s correlation of *Gbp5* and *Ifng* in the colon of WT mice, untreated or treated with DSS. (**E**) Disease activity index (DAI) and (**F**) colon length on day 63 of WT and *Gbp5*^−/−^ mice, treated with DSS, as in (**A**), or not treated (NT). Data are mean ± SEM. n = five per group. NS, not significant; * *p* < 0.05; ** *p* < 0.01; *** *p* < 0.001; Student’s *t*-test. DSS, dextran sulfate sodium.

**Figure 2 biomolecules-14-00873-f002:**
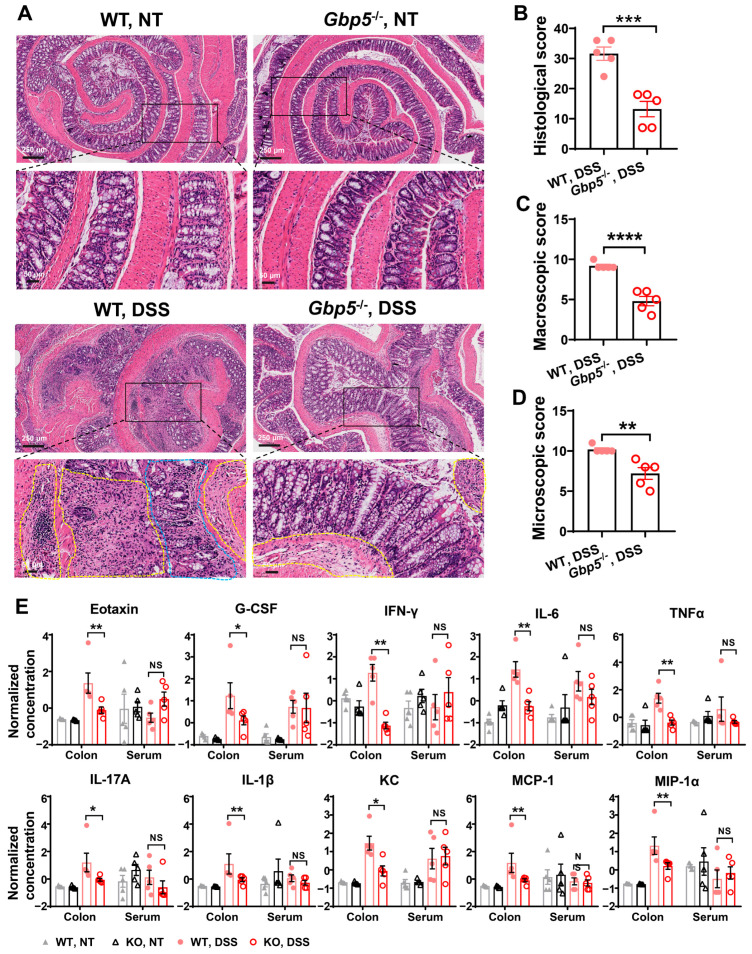
*Gbp5* deficiency alleviates DSS-induced chronic colitis in mice. (**A**) Left: representative hematoxylin and eosin (H&E) stained images of colon cross-sections from WT and *Gbp5*^−/−^ mice, untreated or treated with DSS. The immune cell infiltration in the submucosa and mucosa layers was highlighted with yellow and blue dotted lines, respectively. Scale bars: 250 μm (upper panels); and 50 μm (lower panels). (**B**–**D**) Histopathologic, macroscopic, and microscopic scoring were conducted to examine lymphocyte infiltration and intestinal damage. (**E**) The expression levels of the indicated cytokines in the colons and serum of WT and *Gbp5*^−/−^ mice, untreated or treated with DSS. The concentrations of the cytokines were quantified using a Luminex liquid suspension chip. Data were normalized as (x−mean)/SD. * *p* < 0.05; ** *p* < 0.01; *** *p* < 0.001; **** *p* < 0.0001; Student’s *t*-test.

**Figure 3 biomolecules-14-00873-f003:**
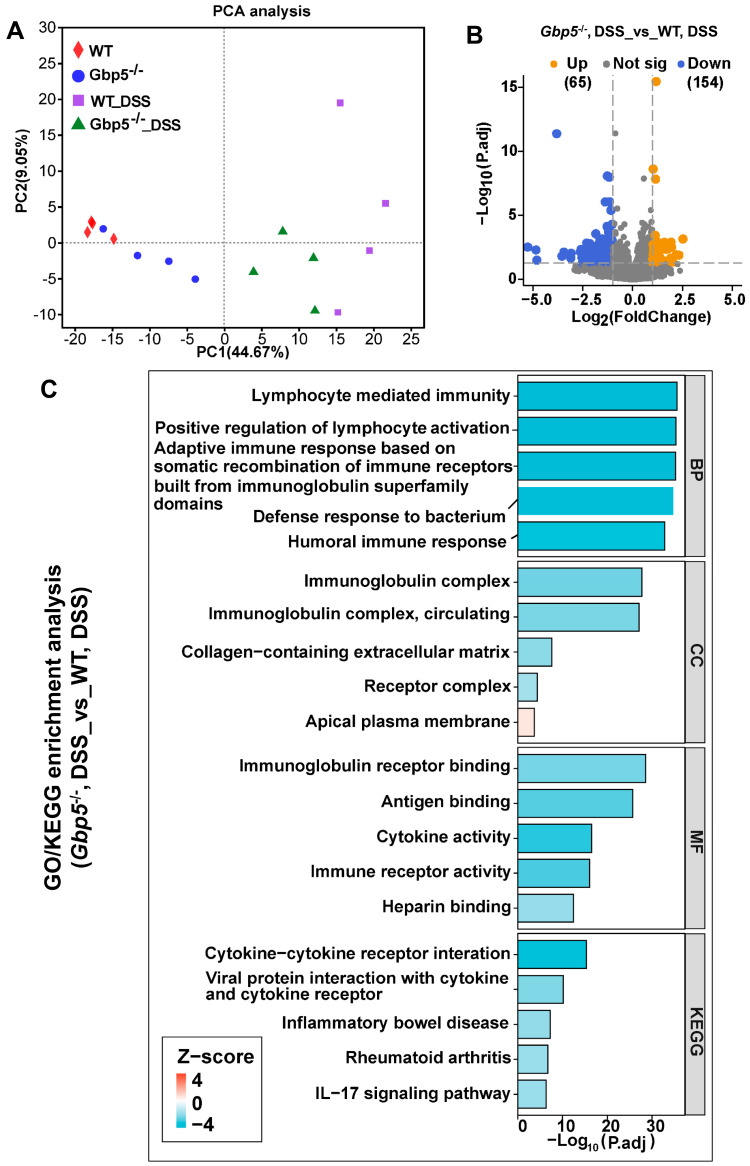
The immune response was downregulated in *Gbp5*^−/−^ mice treated with DSS compared to that of WT mice. (**A**) The colon tissues of mice were subjected to total RNA extraction and RNA sequencing. PCA analysis of transcriptomes from WT and *Gbp5*^−/−^ mice, untreated or treated with DSS. (**B**) Volcano plot showing differentially expressed genes (DEGs) between WT and *Gbp5*^−/−^ mice treated with DSS. (**C**) GO and KEGG enrichment analysis were performed for DEGs. The cut-off values for selecting DEGs: |Log_2_(Fold Change)| ≥ 1 and *p*. adjust < 0.05. BP, biological process; CC, cellular compartment; MF, molecular function.

**Figure 4 biomolecules-14-00873-f004:**
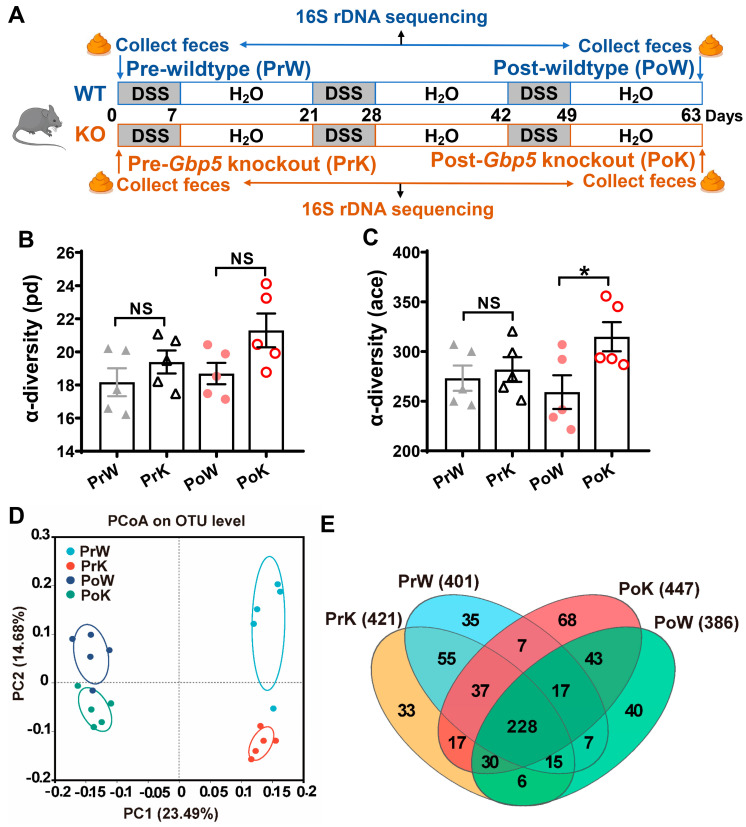
*Gbp5* deficiency induces changes in the community structure of the intestinal flora. (**A**) WT mice and *Gbp5*^−/−^ mice were treated with DSS or water. The DSS cycles were interspersed with 2week periods of plain water administration. Feces collected at day 0 (pre-treatment of WT mice: PrW; pre-treatment of *Gbp5* knockout mice: PrK) and at the end of the experiment (post-treatment of WT mice: PoW; post-treatment of *Gbp5* knockout mice: PoK) were analyzed using 16S ribosomal DNA sequencing. (**B**,**C**) Alpha diversity of microbiome from mice evaluated by phylogenetic diversity (pd) and abundance-based coverage estimator (ace). (**D**) PCoA of microbiotic structure. (**E**) Venn analysis comparing the PrW, PrK, PoW, and PoK groups. NS, not significant; * *p* < 0.05; Student’s *t*-test.

**Figure 5 biomolecules-14-00873-f005:**
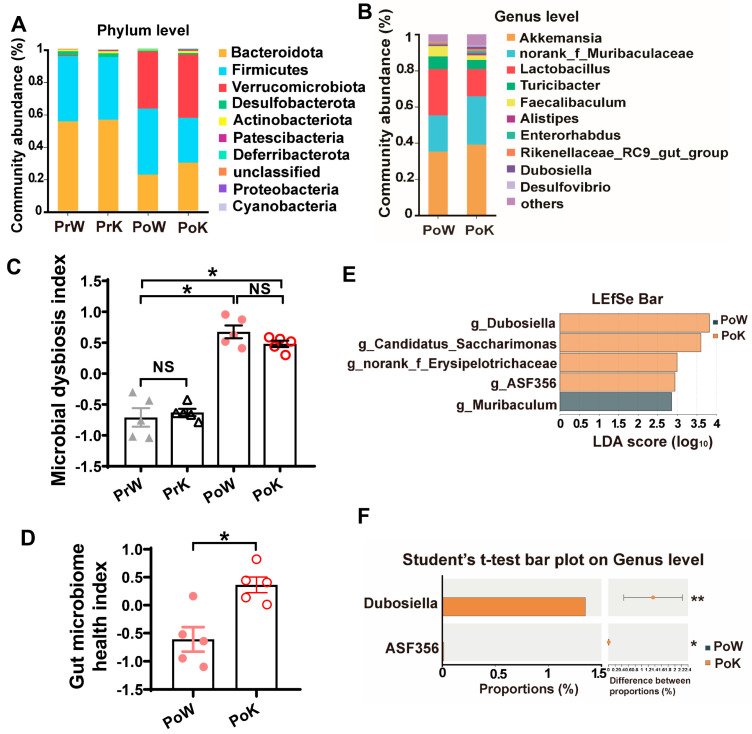
*Gbp5* deficiency promotes gut microbiome health in DSS-induced chronic colitis. (**A**) The relative abundance of the top 10 phyla in the PrW, PrK, PoW, and PoK groups. (**B**) The relative abundance of the top 10 genera in the PoW and PoK groups. (**C**) Microbial dysbiosis index (MDI) in the PrW, PrK, PoW, and PoK groups. (**D**) Gut microbiome health index (GMHI) between PoW and PoK groups. (**E**) Linear discriminate analysis (LDA) at genus levels between PoW and PoK groups by LEFSe. (**F**) Student’s *t*-test at genus levels between PoW and PoK groups. NS, not significant; * *p* < 0.05; ** *p* < 0.01; Student’s *t*-test.

## Data Availability

The samples of RNA-Seq sequencing are available from the NCBI under accession number PRJNA1111593, and 16S rRNA gene sequencing is available from the NCBI under accession number PRJNA1098356.

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
