# Peer review of "Diminished Immune Response and Elevated Abundance in Gut Microbe Dubosiella in Mouse Models of Chronic Colitis with GBP5 Deficiency"

_biomolecules, 2024, doi:10.3390/biom14070873_

Round 1

Reviewer 1 Report (Previous Reviewer 3)

Comments and Suggestions for Authors

The authors addressed alll my comments.

Reviewer 2 Report (Previous Reviewer 2)

Comments and Suggestions for Authors

Authors significantly respond to the reviewers remarks by text edition and additional experiments.

Just a minor remark: First paragraph of the discussion (page 11 of 16, line 369), "alteration of " should be replaced by "alters".

Reviewer 3 Report (Previous Reviewer 1)

Comments and Suggestions for Authors

The authors have incorporated all the reviewers' recommendations in the manuscript's new version. The final version is now more complete.

This manuscript is a resubmission of an earlier submission. The following is a list of the peer review reports and author responses from that submission.

Round 1

Reviewer 1 Report

Comments and Suggestions for Authors

In this work, the authors confirm previous studies of the group already published with an in vivo study of the role of Gbp5 in intestinal inflammation in a DSS-induced colitis mice model using Gbp5 knock-out mice.

They observed lower susceptibility to intestinal inflammation in the Gbp5 knock-out mice using as indicators body weight loss, histological score, and analysis of colon cytokine and immune response pathway RNA levels. The key point of the study is that they observed, following three cycles of DSS, that Gbp5 knock-out mice had different gut microbial composition with better gut microbiome health index than WT mice when both strains of mice had the same microbiota composition before the treatment.  

The scientific background and rationale of the study are well described together with a reference section adequate throughout the text.

MAJOR POINTS

How many experiments have the authors carried out? It seems that they obtained the results based on only one experiment. They obtained the results from only one experiment in which no significant differences in body weight loss between the strains of mice existed. They should repeat it with a high concentration of DSS. Authors should normalize body weight loss to 100% in every DSS cycle. In Figure 1C, when did they calculate the Disease Activity Index (DAI)? Is it a cumulative disease activity index? They should add this information to the figure legend. They should repeat the experiments with a higher concentration of DSS.

MINOR POINTS

To be included into the material and methods section:

-The gender of the mice they have used

-A brief description of the histological score

-A statistics section

They should describe the group of species included in microbial alpha diversity

They should describe what GMHI and LefSe are based on.

Define SCFAs in the discussion section

Author contributions, to be consistent, all names should be the same or with acronyms or the whole name.

Reference numbers 7 and 8 should be more recent.

Reviewer 2 Report

Comments and Suggestions for Authors

In this manuscript, the authors submit Gbp5-/- mice to dextran sodium sulphate (DSS) to induce "chronic" colitis. They observe a mitigation of few colitis markers. Metagenomic analysis revealed Dubosiella enrichment in Gbp5-/- mice after DSS exposure. This study deserves in-depth analyzes of the model used and the omics data. Experiments to explore the link between observations are necessary to support the assertions made by the authors.

Specific comments:

1- Authors used a mouse model of chronic colitis-induced by DSS, a model that combines inflammation, fibrosis and regeneration. It is puzzling and deleterious for the originality of the work that authors only analyze the DAI, the colon length, an histological score (limited to inflammation) and RNA expression of Tnfa and Ifng. These limited data are far to be sufficient to draw conclusion such as "These results showed Gbp5 deficiency protect mice from chronic colitis" (p4 of 14 line 152).

In the same way how could it be concluded "These results highlight the role of GBP5 in promoting inflammation from DSS-induced chronic colitis" by only showing a more limited increase in Tnfa and IFng mRNA level in Gbp5-/- mice receiving DSS compared to WT mice receiving DSS? For such assertion, authors should have treated with GBP5 and deeply characterize inflammation.

- GBP5 expression is inducible by IFNg. How is the Gbp5 expression in gut of WT mice receiving DSS as compared to non DSS-treated controls? Is there a correlation between IFNg and GBP5 in colon ?

- P6 of 14 and Fig3 dealing with transcriptomic analysis highlighted lymphocyte related biological processes and "defense response to bacterium". As it is already established that lymphocytes are the main cellular source of  GBP5 and the fight against bacteria is a known function of the latter (PMID: 22461501), it is hard to catch the novelty of this data. More in depth analysis of these transcriptomic data and subsequent mechanistic experiments are needed.

- Fig 4B & 4C, it is quite surprising that alpha-diversity is not different in WT after DSS (PrW vs PoW). Indeed most studies in the litterature reported a more or less significant decrease in this alpha-diversity in human IBD and in different models such as DSS. Moreover, in the Figure "a-deversity" should be replaced by "a-diversity".

- Paragraph 3.5, authors should not conclude "Gbp5 deficiency induced improvement of colitis may be mediated by the enrichment of Dubosiella". Indeed, at best the results show an enrichment of Dubosiella coincident with an improvement of some markers of colitis in Gbp5-/- mice.

- in the same line, the discussion contains numerous assertions not supported by the data obtained. As an example, the first paragraph (line 260-262), the fact that Gbp5 deficiency reduces DSS susceptibility through immune response and gut microbiota homeostasis was not experimentally proven. Line 267 the acting role of GBP5 in promoting inflammation was not demonstrated in this study. Line 276 where is the experiment that "revealed GBP5 are involved in promoting cytokines expression"? Etc.

Reviewer 3 Report

Comments and Suggestions for Authors

The paper entitled “” presents some introductory data on the influence of GBP5 on the occurrence and development of colonic inflammation in vivo. The authors perform microbiome analysis in wild type and knockout mice subjected to DSS-induced colon inflammation and studied its influence on inflammatory reaction and microbiota. This study is interesting and show GBP5 protein modulated inflammation within the colon tissue but it could be improved.

Results:

To support gathered results and strengthen conclusion the authors should include colon macroscopic and microscopic scoring like e.g. in doi: 10.1007/s43440-020-00190-3

The authors have to add statistical analysis description.

What was the annealing temperature for primers?

Mice:

The authors write that “Gbp5-/- mice in a C57BL/6J”, and “Mice were bred and maintained under specific pathogen free (SPF) facilities at the Sixth Affiliated Hospital of Sun Yat-Sen University. ”. Hence the animal facility breads the mice and mice they were another strain and should have other nomenclature: C57BL/6J/(shortcut for Hospital of Sun Yat-Sen University animal facility).

How many mice was in each group and which age were there?

Why the authors excluded female mice?

Did the authors perform any functional test of GI function and/or microbiome transfer? If not then why?

How disease activity index was calculated?

The mice were bread and hosted in SPF condition which strongly influence gut microbiota, the result gathered in conventional conditions could be different. The authors should comment on that in the discussion.

Results:

The authors should add en face picture of colon with cecum that supports results presented in Figure 1D.

The authors should present data protein level of tested inflammatory parameters in plasma and colon (e.g. with ELISA or western blot). Simple analysis of RNA level is not sufficient as mRNA expression is changing rapidly.

It would be valuable if the authors present inflammatory cells infiltration within the colon.

Conclusion;

The authors state that “These study exhibit Dubosiella may represent a promising therapeutic target for diseases 305 with dysbiosis.”. How significant and true this statement could be for mice treated in conventional conditions?